# Microarray Gene Expression Analysis of Lesional Skin in Canine Pemphigus Foliaceus

**DOI:** 10.3390/vetsci11020089

**Published:** 2024-02-14

**Authors:** Haley Starr, Elizabeth W. Howerth, Renato Leon, Robert M. Gogal, Frane Banovic

**Affiliations:** 1Department of Small Animal Medicine and Surgery, College of Veterinary Medicine, University of Georgia, Athens, GA 30602, USA; haley.starr@uga.edu (H.S.); rgl84554@uga.edu (R.L.); 2Department of Pathology, College of Veterinary Medicine, University of Georgia, Athens, GA 30602, USA; howerth@uga.edu; 3Department of Biomedical Sciences, College of Veterinary Medicine, University of Georgia, Athens, GA 30602, USA; rgogal@uga.edu

**Keywords:** canine, pemphigus foliaceus, RNA microarray

## Abstract

**Simple Summary:**

Canine pemphigus foliaceus (PF) is considered the most common autoimmune skin disease in dogs. The mechanisms of canine PF disease are currently poorly understood; therefore, this study aimed to better understand the immune signature of canine PF. We analyzed the expression of 800 different genes from formalin-fixed, paraffin-embedded samples. Our transcriptome analyses found 420 significantly differentially expressed genes (DEGs, 338 upregulated and 82 downregulated). These genes belonged to several signaling pathways, such as the mitogen-activated protein kinase (MAPK), the Janus kinase signal transducer and activator of transcription (JAK-STAT), and T-helper 17 (Th17) pathways, which are also found to be enhanced in human pemphigus studies and could provide a novel target for future drug therapy.

**Abstract:**

Canine pemphigus foliaceus (PF) is considered the most common autoimmune skin disease in dogs; the mechanism of PF disease development is currently poorly understood. Therefore, this study aimed to characterize the molecular mechanisms and altered biological pathways in the skin lesions of canine PF patients. Using an RNA microarray on formalin-fixed, paraffin-embedded samples, we analyzed the transcriptome of canine PF lesional skin (*n* = 7) compared to healthy skin (*n* = 5). Of the 800 genes analyzed, 420 differentially expressed genes (DEGs) (*p* < 0.05) were found. Of those, 338 genes were significantly upregulated, including pro-inflammatory and Th17-related genes. Cell type profiling found enhancement of several cell types, such as neutrophils, T-cells, and macrophages, in PF skin compared to healthy skin. Enrichment analyses of the upregulated DEGs resulted in 78 statistically significant process networks (FDR < 0.05), including the Janus kinase signal transducer and activator of transcription (JAK-STAT) and mitogen-activated protein kinase (MAPK) signaling. In conclusion, canine PF lesional immune signature resembles previously published changes in human pemphigus skin lesions. Further studies with canine PF lesional skin using next-generation sequencing (e.g., RNA sequencing, spatial transcriptomics, etc.) and the development of canine keratinocyte/skin explant PF models are needed to elucidate the pathogenesis of this debilitating disease.

## 1. Introduction

Pemphigus is a group of bullous autoimmune diseases affecting many species, including humans and dogs [1]. The pemphigus diseases are characterized by autoantibodies that target desmosomal adhesion molecules (e.g., desmogleins and desmocollins), resulting in keratinocyte acantholysis, inflammatory response, blister formation, and erosions [2]. The specific autoantibody target defines several subtypes of pemphigus; pemphigus vulgaris (PV), the most common human pemphigus form, targets the deep layers of the epidermis, whereas pemphigus foliaceus (PF) represents the most common canine pemphigus form, which affects the uppermost layers of the epidermis, forming superficial pustules and crusts [1,2]. 

Several studies have focused on elucidating the pathogenesis and mechanisms underlying the loss of cell adhesion in human pemphigus skin lesions using gene expression analysis, murine models, and cultured keratinocyte pemphigus models [3,4]. Historically, it was assumed that autoantibody-mediated diseases like pemphigus were driven by classical T helper 2 (Th2) cells producing the lineage-characteristic cytokine interleukin 4 (IL-4). The development of keratinocyte dissociation pemphigus models allowed for the identification of several signaling pathways (e.g., p38 mitogen-activated protein kinase (MAPK), phospholipase C (PLC)/protein kinase C (PKC) activation), with different steps of desmosome turnover after pemphigus autoantibodies engage desmosomal targets [5,6]. A recent study identified a prominent T helper (Th) 17 pathway with elevated Janus kinase (JAK)-signal transducer and activator of transcription (STAT) signaling in human pemphigus lesions and keratinocyte pemphigus models, opening a potential avenue for investigating novel targeted therapeutics [3]. 

In contrast to human pemphigus research, the pathogenesis of canine PF is poorly investigated, with studies predominantly focused on uncovering the major autoantibody target (e.g., desmocollin-1) [7,8]. A recent gene expression study using quantitative reverse-transcription PCR (qRT-PCR) analysis with a limited number of immune markers showed upregulation of interleukin (IL)-8, interferon-gamma (IFNG), IL-22, IL-17, and IL-13 [9]. 

Considering the current lack of knowledge regarding the pathogenesis of canine pemphigus and the impact and burden of such a debilitating disease upon affected patients and owners, there is a clear need to further the understanding of the mechanisms of canine PF skin lesion development. In this study, we analyzed the skin lesional transcriptome of seven dogs affected by PF using an RNA microarray technique with 780 genes; healthy skin from five healthy dogs was used as the control.

## 2. Materials and Methods

### 2.1. Patient Inclusion

All study protocols were reviewed, approved, and conducted per the Institutional Animal Care and Use Committee (IACUC Protocol CR-509). After owners signed the informed consent form, all dogs included in this study were prospectively enrolled at the Veterinary Teaching Hospital’s Dermatology service. The transcriptome analysis in this study was performed using lesional skin samples from seven patients with canine PF, whereas skin tissues from five healthy dogs served as a control. A minimum number of 5 samples per group was determined to be sufficient to achieve 80% statistical power to detect a significant effect (*p* < 0.05), estimating a difference in at least 400 genes in diseased skin compared to healthy skin [10].

Canine PF patients were diagnosed based on previously published criteria of history, clinical presentation, and microscopic findings (cytological and histopathological) of acantholytic keratinocytes without the presence of bacteria or fungal organisms [11,12,13]. Tissues from five privately owned, healthy dogs were utilized as controls; these dogs had no history and showed no evidence of recent or chronic medical conditions and had no abnormalities upon physical examination, complete blood count, or serum biochemical evaluation. The withdrawal times implemented for medications to limit the influence of any drugs on mRNA expression in skin tissues of PF and healthy dogs were two weeks for antihistamines, four weeks for oral/topical/ear glucocorticoids or any other systemic immunomodulating medications (cyclosporine, mycophenolate mofetil, oclacitinib, etc.), and six weeks for injectable glucocorticoids.

### 2.2. Tissue Samples and RNA Extraction

All dogs were sedated with dexmedetomidine hydrochloride (5 μg/kg, Dexdomitor, Zoetis, Florham Park, NJ, USA), and subcutaneous injection of 0.5 mL of lidocaine hydrochloride 2% (Hospira Inc., Lake Forest, IL, USA) was administered at the site of the biopsy to provide additional local anesthesia. One 8 mm skin biopsy sample was obtained from lesional pemphigus areas (e.g., pustules) on the ventral abdomen of PF dogs; identical site-matched samples were taken from healthy dogs as a control. Collected tissues were immediately placed in 10% neutral buffered formalin for paraffin embedding and routinely processed for histopathological examination by a board-certified pathologist (Howerth). 

RNA was extracted from 10 µm thick formalin-fixed, paraffin-embedded (FFPE) tissue sections according to the manufacturer’s instructions using the RNeasy FFPE kit (Qiagen, Hilden, Germany). Briefly, razor blades treated with RNase were used to cut away excess paraffin and slice samples into 5 µm sections. For every sample, a new blade was used to avoid any potential contamination. The 5 µm sections were placed in a deparaffination solution (Qiagen, Hilden, Germany) before quantifying the RNA concentration using the Nanodrop 2000 Spectrophotometer (ThermoFisher Scientific, Waltham, MA, USA). DV200 values were obtained using the Bioanalyzer 2100 (Agilent Technologies, Santa Clara, CA, USA). All samples were required to have DV200 > 70% to be included in further analysis.

### 2.3. Microarray Analysis Using the NanoString Canine IO Panel

The extracted FFPE RNA was hybridized with a gene-specific reporter and capture probes and processed using the Nanostring nCounter analysis system (nCounter Canine Immuno-Oncology (IO) Panel, NanoString, Seattle, WA, USA). Gene expression was quantified using the Nanostring Canine IO Panel, which consists of 780 genes, including 305 cytokine and chemokine signaling genes, 47 interferon signaling genes, 67 checkpoint signaling genes, 52 genes related to immune cell abundance, 74 tumor immunogenicity genes (e.g., antigen presentation and DNA damage repair), 94 genes of inhibitory tumor mechanisms (e.g., epigenetic regulation, hypoxia, transforming growth factor-beta (TGF-B) signaling, Wnt signaling), and 101 genes associated with stromal factors (angiogenesis and matrix remodeling and metastasis).

### 2.4. Differential Expression Analysis and Cell Type Profiling

Data were analyzed using ROSALIND software (version 3.16; URL https://rosalind.bio/; accessed on 1 December 2023; San Diego, CA, USA). Housekeeping probes for normalization were selected based on the geNorm algorithm [14]. Normalization, fold changes (FC), and adjusted *p*-values (*p*-adj) were calculated as described in the nCounter Advanced Analysis 2.0 User Manual by dividing counts within a lane by the geometric mean of the normalizer probes from the same lane. Differentially expressed genes (DEGs) were defined as all genes with a *p*-adj of less than 0.05.

### 2.5. Cell Type Profiling

The abundance of various cell populations between healthy and pemphigus lesional skin was analyzed using nSolver software (version 2.0.134, NanoString Cell Type Profiling Module; NanoString) and GraphPad Prism software (version 9.5, GraphPad Software, San Diego, CA, USA). Data were initially evaluated for distribution, and the two-tailed Mann–Whitney U-test was used for non-normally distributed data. Only cell type scores with a *p*-value less than or equal to 0.05 were considered significant.

### 2.6. Enrichment Analysis

The process network tool from MetaCore software (Version v2.0.4, Clarivate Analytics, London, UK) was utilized for the functional annotation of the DEGs and to identify significantly enriched pathways regarding the hypergeometric distribution based on a false discovery rate (FDR) <0.05.

## 3. Results

### 3.1. Patient Characteristics and Histopathological Examination

The signalments for dogs affected by PF and healthy dogs are provided in Table 1.

Microscopically, acantholytic keratinocytes and a hyperkeratotic epidermis were noted in all biopsies from canine PF patients; no bacterial or fungal organisms were observed per the inclusion criteria. Acantholytic keratinocytes were within a superficial pustule or seen as keratinocyte ghosts within a serocellular crust, both spanning multiple hair follicles (three of seven). Heavy neutrophilic infiltration of the epidermis was detected in six PF patients (six of seven), whereas heavy eosinophilic infiltration with fewer neutrophils was present in the dermis of a single PF dog. Plasma cells, lymphocytes, and mast cells were found in overall smaller amounts within the dermis of most PF patients (six of seven). The presence of lymphocytes and plasma cells without mast cells was noted in a single sample (one of seven).

### 3.2. Differential Gene Expression (DEGs) Analysis

From the 780 total genes analyzed, 420 genes were statistically significantly differentially expressed (*p*-adj < 0.05; Appendix A). Overall, there were 338 significantly upregulated and 82 significantly downregulated markers. A principal component analysis (PCA, Appendix A) demonstrated a clear separation of the PF and healthy skin samples.

The pro-inflammatory (IL-1 beta B), IL-8/CXCL8, and Th17 (IL-17A, IL-17 F, IL-19, S100 calcium-binding protein (S100) A9) markers showed the highest expressions among the top 10 upregulated genes in lesional canine PF skin. The top 10 downregulated genes involved CC motif chemokine ligand (CCL) 27 (FC: 3.8; *p*-adj < 0.0005), S100B (FC: 2.3; *p*-adj < 0.01), GATA3 (FC: 2.3; *p*-adj < 0.0001), estrogen receptor (ESR) 1 (FC: 2.2; *p*-adj < 0.0005), complement factor (CF) D (FC: 2.0; *p*-adj < 0.0001), cluster of differentiation (CD) 34 (FC: 1.7; *p*-adj < 0.005), AR (FC: 1.6; *p*-adj < 0.0005), CCL21 (FC: 1.6; *p*-adj < 0.005), thioredoxin-interacting protein (TXNIP) (FC: 1.5; *p*-adj < 0.005), and fibroblast growth factor receptor 1 (FGFR1) (FC: 1.4; *p*-adj < 0.0001).

Canine lesional PF skin featured prominent immune changes (Figure 1) for several pro-inflammatory genes (tumor necrosis factor-alpha (TNF-a), IL-1A, IL-1B, IL-6, IL-18, and IL-8) and Th17-related markers (IL-17A, IL-17F, IL-17RA, IL-23A, IL-23R, and STAT3). In addition, upregulation of Th1 markers (IL-2RA, interferon regulatory factor 1 (IRF1), STAT4, and interferon-gamma receptor 1 (IFNGR1)), several Th2 genes (IL-4R, IL-13RA1, IL-13RA2, and oncostatin M (OSM)), and Th22 marker S100A9 was also found. 

Among the chemokines and receptors (Figure 1), a robust upregulation was found for CCL2, CCL3, CCL4, CCL8, CCL17, CCL20, CCL23, and CCL28 as well as chemokine (C-X-C motif) ligand (CXCL)10, whereas CXCR1, the central neutrophil receptor for trafficking, was the most upregulated chemokine receptor. 

Genes associated with the FAS cell signaling pathway, including caspase 10 (CASP10; FC: 1.3 *p*-adj < 0.001), CASP3 (FC: 0.60, *p*-adj < 0.001), CASP8 (FC: 1.5, *p*-adj < 0.01), FAS associated via death domain (FADD) (FC: 0.52 *p*-adj < 0.01), and FAS (FC: 2.6, *p*-adj < 0.0001), were also found to be significantly differentially expressed in canine PF skin compared to healthy controls.

### 3.3. Cell Type Profiling

Of the nine different cell types analyzed (Figure 2), CD45 cells (*p* < 0.0001), natural killer (NK)/CD56^dim^ cells (*p* < 0.0001), T-cells (*p* < 0.005), neutrophils (*p* < 0.0001), dendritic cells (*p* < 0.0005), and CD8 T cells (*p* < 0.05) were significantly more abundant in canine PF diseased skin lesions compared to healthy controls. There was no statistically significant difference between healthy and diseased skin when comparing the abundance of mast cells, cytotoxic cells, or exhausted CD8 cells.

### 3.4. Enrichment Analysis Using Metacore Process Networks

To understand whether the DEGs are significantly enriched in the transcriptional enrichment and pathway analysis, upregulated and downregulated DEGs were analyzed using Metacore software. In our study, canine PF skin lesions showed 78 upregulated and 74 downregulated process networks (Appendix A). 

The top 10 upregulated pathways were associated with lymphocyte proliferation (66 from a total of 208 genes in the pathway), triggering receptor expressed on myeloid cells (TREM) 1 signaling (51/145), JAK-STAT (56/185), and IL-4 signaling (45/115). In addition, other upregulated pathways included Th17 derived cytokines (34/98), NK cell cytotoxicity (35/164), anti-apoptosis mediated by external signals via MAPK and JAK/STAT (28/179), anti-apoptosis stimulation by external signals via nuclear factor-kappa beta (NF-kB) (26/111), nitric oxide signaling (20/88), death domain receptors and caspases in apoptosis (20/123), role of reduced nicotinamide adenine dinucleotide phosphate (NADPH) oxidase and reactive oxygen species (ROS) (17/134), ESR1-nuclear pathway (17/217), ERbB family signaling (15/75), response to hypoxia and oxidative stress (11/161), and ESR1-membrane pathway (10/91).

For the downregulated genes, the top 10 process networks included NOTCH signaling (25/235), the B cell receptor (BCR) pathway (15/137), IL-2 signaling (12/104), anti-apoptosis by external signals via phosphoinositide-3-kinase/protein kinase B (P13K/AKT) (14/233), the ESR1 membrane pathway (13/91), T cell receptor (TCR) signaling (13/74), and fibroblast growth factor (FGF)/ErbB signaling (12/124).

## 4. Discussion

In the present study, we aimed to analyze the molecular mechanisms and altered biological pathways in the skin lesions of canine PF patients. Previous studies evaluating immune markers of canine PF patients are limited to two reports [9,15]. In the first study, the Nanostring microarray with 160 markers was used to analyze lesional skin gene expressions from a single PF patient [15], while in the second study, qRT-PCR was used in seven canine PF skin tissues to evaluate changes in only 20 immune markers [9]. This study identified 420 DEGs possibly involved in the pathogenesis of canine PF skin lesion development. 

Interestingly, our transcriptome results showed IL-8, IL-19, IL-17A, IL-17F, IL-1B, S100A9, S100A12, S100A8, and IL-23A to be the top upregulated immune markers. Furthermore, cell profiling enrichment revealed the activation of T cells, CD45 cells, and neutrophils in lesional canine PF samples; the activation of these types of cells corresponds to the histopathological features of canine PF and has been previously reported in human and canine pemphigus [15,16,17]. Several chemokines and chemokine receptors (i.e., IL-8 (CXCL8), CCL3, CCL4, CCL8, CCL20, CCL28, and CXCL10) that could contribute to the immune infiltration of immune cells in skin lesions displayed strong gene expression upregulation in canine PF lesions. Interleukin-8 is considered the main recruiter of neutrophils in the epidermis, promoting PF blister formation [18], whereas CXCL10 may play a role in the induction of matrix metalloproteinase 9 (MMP-9), a proteinase that contributes to degradation in blister formation [19].

Pemphigus was initially considered a classic Th2 disease due to the production and presence of circulating autoantibodies by B cells. Furthermore, IL-4 was considered the main cytokine for promoting isotype switching and inducing naïve CD4+ T cells to differentiate into Th2 cells. A recent study that evaluated the skin cytokine signature and immunophenotyped peripheral blood cells demonstrated an IL-17-dominant signature in both skin lesions and the peripheral blood of human PV and PF patients [3]. In line with previous human pemphigus studies [3,20,21,22], the data presented here confirm the overexpression of IL-17A, IL-17F, IL-17RA, IL-23A, and IL-23R in canine PF skin.

Interlukin-17 is a pro-inflammatory cytokine with several functions, including stimulation of neutrophils and production of TNF, IL-1α, IL-1β, and IL-8 [23]. The pro-inflammatory Th17 pathway has been suggested to play a significant role in the pathogenesis of several chronic inflammatory and autoimmune skin diseases, such as psoriasis, atopic dermatitis, dermatitis herpetiformis, and hidradenitis suppurativa [24,25,26]. Interestingly, anti-IL-17A monoclonal antibody secukinumab significantly improved the skin lesions and circulating levels of anti-desmoglein (Dsg) 1 antibodies in a human PF patient [27]. Further studies will be needed to evaluate whether inhibitors of the IL-17 pathway (e.g., IL-17A, IL-17A/IL-17F, and IL-23A) could be used to treat the pemphigus group of blistering diseases in humans and dogs.

In recent years, several JAK inhibitors have been developed for the treatment of inflammatory skin diseases in humans and dogs [28,29,30,31]. Several studies suggest the potential involvement of the JAK-STAT functional pathway in human pemphigus via the following mechanisms: (i) overexpression of JAK-signaling cytokines (e.g., IL-6 and IL-21) important for autoantibody production in pemphigus, and (ii) JAK1/JAK3 inhibitors demonstrate a protective effect on keratinocyte cell acantholysis after incubation with pemphigus autoantibodies, suggesting a potential involvement of JAK/STAT molecules during blister formation in pemphigus disease [3]. Similarly to the human pemphigus data, the JAK-STAT pathway was among the top 10 upregulated enrichment pathways in canine lesional PF skin in this study. While clinical efficacy studies with JAK inhibitors for human pemphigus are still lacking, oral oclacitinib, a veterinary-registered JAK inhibitor, has been successfully used in open-label reports to treat canine and feline pemphigus patients [29,32]. 

Several intracellular signals (e.g., p38MAPK signaling) that could lead to keratinocyte apoptosis have been investigated after binding of human pemphigus autoantibodies to desmosomal targets. A recent study revealed that tubules of endoplasmic reticulum (ER) are associated with desmosomes, and ER stress, a known activator of p38 MAPK pathway [33], develops with any type of desmosome or keratin filament perturbation [34]. Our canine PF lesional skin data showed upregulation of several p38 MAPK pathway genes, such as MAPK14, MAP2K4, MAPKAPK2, STAT4, IL-1R1, and tumor necrosis factor receptor-associated factor 6 (TRAF6). Treatment with p38MAPK inhibitors in human pemphigus has been limited by severe side effects (e.g., hepatotoxicity and gastrointestinal toxicity) before efficacy could be evaluated [35]; to the best of the authors’ knowledge, there have been no clinical studies using p38MAPK inhibitors in dogs. Our study also found upregulation of genes associated with the FAS pathway (e.g., FAS, FADD, CASP10, CASP3, and CASP8). Several reports investigated the relevance of the FAS pathway in human pemphigus blister formation; upregulation of FAS ligand after pemphigus autoantibody binding induces keratinocyte apoptosis and desmoglein cleavage through the activation of caspase-8 [36,37]. PC111, a human monoclonal antibody targeting the FAS ligand, significantly reduced acantholysis using a keratinocyte dissociation and human skin model after treatment with pemphigus autoantibodies [38]. 

## 5. Conclusions

In summary, we found robust immune responses in the skin lesions of canine PF. This included a strong upregulation of pro-inflammatory and IL17-related cytokines, which aligns with previously published data from human pemphigus skin lesions. In addition, we showed activation of the JAK-STAT pathway in lesional skin, which could represent a novel target for the treatment of canine PF. Several limitations of our study include a small sample size, a limited number of genes in the microarray design, validation of the mRNA results through protein expression studies, and the time variability of the PF lesion development. Although we did not observe specific molecular differences, future PF studies may consider evaluating the differences between histological neutrophilic and eosinophilic PF groups. Further studies with canine PF lesional skin using next-generation sequencing (e.g., RNA sequencing, spatial transcriptomics, etc.) and the development of canine keratinocyte/skin explant PF models are needed to elucidate the pathogenesis of this debilitating disease.

## Figures and Tables

**Figure 1 vetsci-11-00089-f001:**
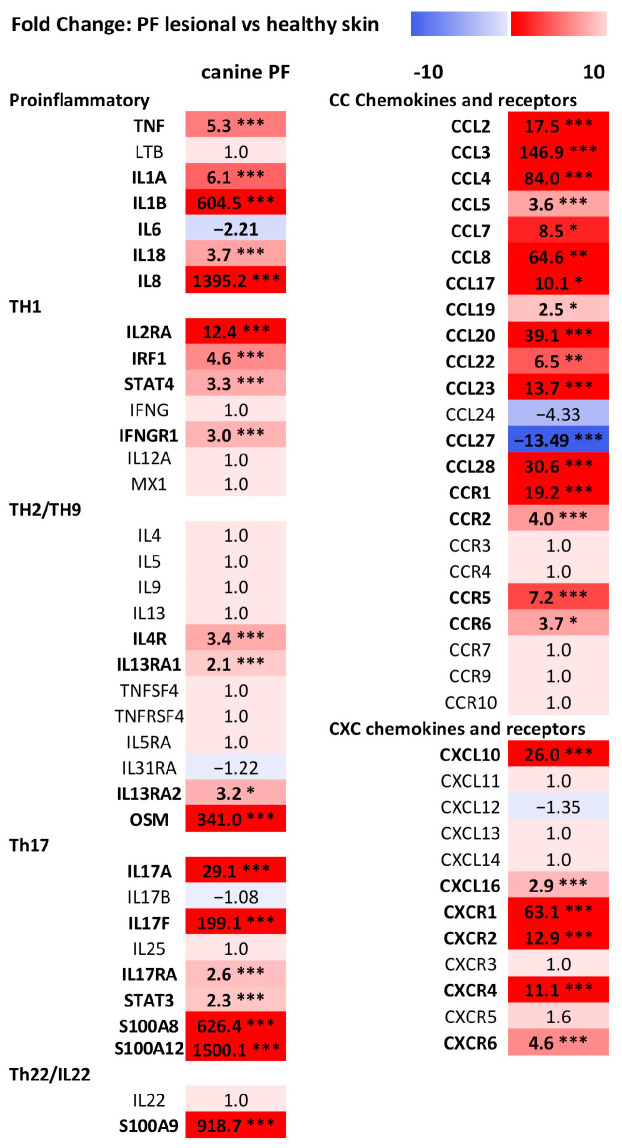
Expression of selected relevant cytokine, chemokine, and chemokine receptor genes in lesional pemphigus foliaceus (PF) skin (*n* = 7) compared to healthy skin (*n* = 5). Genes are arranged according to their dominant function or family, and the color responds to downregulation (dark blue) and upregulation (bright red); bolded fold changes (FC) with asterisks are statistically significant at * *p*-adj < 0.05, ** *p*-adj < 0.01, *** *p*-adj < 0.005.

**Figure 2 vetsci-11-00089-f002:**
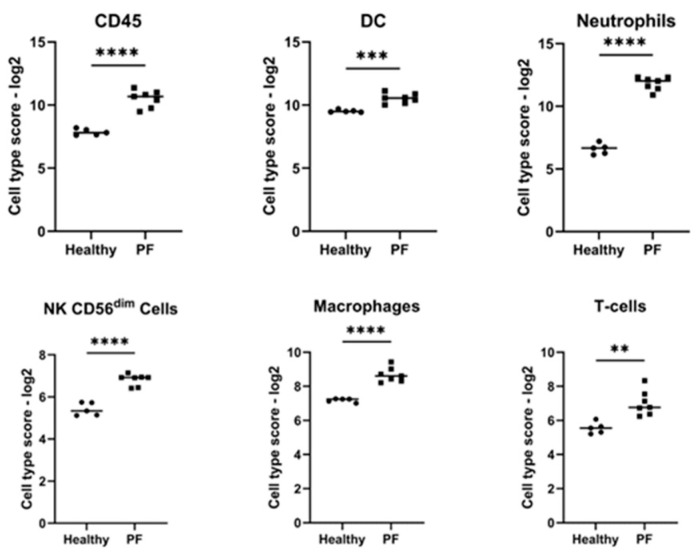
Scatter plots with mean values (solid horizontal bar) showing significant log2 cell type scores of different cell types in 7 dogs affected by pemphigus foliaceus (PF) compared to 5 healthy controls. Cell types include dendritic cells (DCs), CD45 cells, neutrophils, CD8 T-cells, NK 56 dim cells, and T-cells. Significance is indicated by asterisks at ** *p* < 0.01, *** *p* < 0.005, **** *p* < 0.001. Data are presented as individual samples with medians.

**Table 1 vetsci-11-00089-t001:** Signalments and historical features of 7 dogs with pemphigus foliaceus and 5 healthy control dogs that were used for transcriptome analysis using an RNA microarray.

Group	Breed	Age (yr)	Sex	Pemphigus Lesion Distribution	Prior Skin Disease	Prior Non-Cutaneous Disease
Healthy	German Shepard Mix	8.6	FS	None	None	None
	American Staffordshire Terrier	5.6	M	None	None	None
	Mixed	3	MC	None	None	None
	Golden Retriever	1.8	MC	None	None	None
	German Shepard	2.2	FS	None	None	None
PF	Beagle	4.6	MC	Facial and Truncal	None	None
	Boxer Mix	7.8	FS	Facial and Truncal	None	None
	Mixed	7	FS	Facial and Truncal	None	None
	Catahoula Mix	0.75	MC	Facial and Truncal	None	None
	Labrador Retriever	7	FS	Facial and Truncal	None	None
	Pitbull Mix	0.75	FS	Facial and Truncal	None	None
	Mixed	6	FS	Facial and Truncal	None	None

PF, pemphigus foliaceus; MC, male castrated; FS, female spayed.

## Data Availability

Data are contained within the article and Appendix A. The raw sequencing data are available upon request.

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
