# Peer review of "Microarray Gene Expression Analysis of Lesional Skin in Canine Pemphigus Foliaceus"

_vetsci, 2024, doi:10.3390/vetsci11020089_

Round 1

Reviewer 1 Report

Comments and Suggestions for Authors

Dear Authors,

I completed my review on the manuscript entitled “Microarray gene expression analysis of lesional skin in canine pemphigus foliaceus” by Starr et al.

I appreciate that this article is very well done: the study design is well conducted, the conceptualization is flawless and data valorisation is greatly performed. 

It is my opinion that this article is totally acceptable for publication.

However, I do have one minor observation/suggestion:

Section 2. Materials and Methods 2.1. Patient inclusion:

-       please specify whether the “microscopic findings of acantholytic keratinocytes” is the result of a histological or cytological exam.

-   “…without presence of fungal organisms” – was this aspect assessed by fungal culture or by other diagnostic techniques. Please specify.

Reviewer 2 Report

Comments and Suggestions for Authors

Dear Authors,

This paper is extremely interesting for people interested in veterinary dermatology. Please clarify these few points.

line 87: “and microscopic findings of acantholytic keratinocytes without the 86 presence of bacteria or fungal organisms [11-13].” it should be added the exclusion of Leishmaniasis (PF and Leishmania), just because it is in the diseases that in the last years have to be excluded.

line 94-95: and six weeks for injectable glucocorticoids. (is it possible to know what kind of steroids?

table 1: sorry for this question but 0.75?? what does it mean? seven and half months? quite improbable for this disease.

Line 154-155: “no bacterial or fungal organisms were observed per inclusion criteria.” see comment for line 87.

line 305-307: “Several limitations of our study include a small sample size, a limited number of genes in the microarray design, validation of the mRNA results through protein expression studies, and the time-variability of the PF lesion development”. It should be added into the discussion the there is a difference of expression of the studied genes and inflammatory markers among the different type on PF histological lesions (e.g. “Heavy neutrophilic infiltration of the epidermis was detected in six PF patients (6 of 7), whereas heavy eosinophilic infiltration with fewer neutrophils was present in the dermis of a single PF dog. Plasma cells, lymphocytes, and mast cells were found in overall smaller amounts within the dermis of most PF patients (6 of 7). The presence of lymphocytes and plasma cells without mast cells was noted in a single sample (1 of 7)). lines 153-161.
